# Effects of Closed Mouth vs. Exposed Teeth on Facial Expression Processing: An ERP Study

**DOI:** 10.3390/bs15020163

**Published:** 2025-02-01

**Authors:** Nicolas M. Brunet, Alexandra R. Ackerman

**Affiliations:** Department of Psychology, California State University of San Bernardino, San Bernardino, CA 92407, USA; alexandrar1ack@gmail.com

**Keywords:** ERP, EEG, P100, N170, emotional faces, face processing, open and closed mouth, exposed teeth

## Abstract

The current study examines the neural mechanisms underlying facial recognition, focusing on how emotional expression and mouth display modulate event-related potential (ERP) waveforms. 42 participants categorized faces by gender in one of two experimental setups: one featuring full-face images and another with cropped faces presented against neutral gray backgrounds. The stimuli included 288 images balanced across gender, race/ethnicity, emotional expression (“Fearful”, “Happy”, “Neutral”), and mouth display (“closed mouth” vs. “open mouth with exposed teeth”). Results revealed that N170 amplitude was significantly greater for open-mouth (exposed teeth) conditions (*p* < 0.01), independent of emotional expression, and no interaction between emotional expression and mouth display was found. However, the P100 amplitude exhibited a significant interaction between these variables (*p* < 0.05). Monte Carlo simulations analyzing N170 latency differences showed that fearful faces elicited a faster response than happy and neutral faces, with a 2 ms delay unlikely to occur by chance (*p* < 0.01). While these findings challenge prior research suggesting that N170 is directly influenced by emotional expression, they also highlight the potential role of emotional intensity as an alternative explanation. This underscores the importance of further studies to disentangle these effects. This study highlights the critical need to control for mouth display when investigating emotional face processing. The results not only refine our understanding of the neural dynamics of face perception but also confirm that the brain processes fearful expressions more rapidly than happy or neutral ones. These insights offer valuable methodological considerations for future neuroimaging research on emotion perception.

## 1. Introduction

Face and emotion recognition are essential for social interaction, communication, and building relationships ([2]). These abilities enable the interpretation of emotions, fostering empathy, trust, and effective decision-making, which are vital for navigating social contexts and maintaining societal cohesion. Deficits in these skills can arise from developmental disorders such as Autism Spectrum Disorder ([51]), neurological conditions such as Alzheimer’s ([46]), strokes ([52]), or brain injuries ([39]), as well as mental health issues including depression, schizophrenia ([13]), and social anxiety ([47]) disorders. Such impairments can lead to misinterpreting social cues, resulting in isolation, strained relationships, poor communication, heightened anxiety, depression, and stress ([13]; [45]). Misreading emotions may also provoke inappropriate or aggressive responses ([30]). Advancing our understanding of these deficits can pave the way for early interventions and treatments, enhancing social engagement and improving quality of life.

### 1.1. Neurobiology of Facial Processing

Facial recognition involves processing a wealth of information embedded in faces, including age, gender, emotional expression, and subjective traits such as attractiveness, trustworthiness, and competence. This complex task is supported by an intricate neural network specialized for face processing ([14]), with 25 distinct brain regions identified by Zhen et al. as reliably selective for faces ([53]). Key areas in this network include the Fusiform Face Area (FFA) ([27]), located in the fusiform gyrus, which is crucial for recognizing faces and differentiating them from other objects. Damage to the FFA is linked to prosopagnosia, or face blindness. Another critical region is the Occipital Face Area (OFA) ([37]), which processes early-stage facial features such as the eyes, nose, and mouth. The Superior Temporal Sulcus (STS) ([26]) plays a role in interpreting dynamic facial cues, such as gaze direction, and is also involved in the perception of emotional expressions. The amygdala is essential for recognizing emotional expressions, particularly those related to fear and threat ([22]; [49]).

Facial processing relies on two complementary visual pathways ([3]). The dorsal pathway processes the spatial and dynamic aspects of faces, such as gaze direction and movement, while the ventral pathway focuses on the detailed analysis of facial features and identity. Together, these regions and pathways form a sophisticated network that underpins our ability to recognize and interpret faces in social contexts.

### 1.2. Temporal Dynamics

Electroencephalography (EEG), and specifically Event-Related Potentials (ERPs), are commonly used to study the temporal dynamics of face processing due to their high temporal resolution. ERPs are time-locked brainwave responses to specific stimuli, such as faces, allowing researchers to track neural activity at precise moments after stimulus presentation. Several key components have been identified that correspond to distinct stages of facial perception, including the P100, N170, and P300. The P100, occurring around 100 milliseconds after face presentation, marks an early stage of facial perception, typically measured over occipital electrodes and associated with OFA ([42]). It reflects the initial visual processing of a face stimulus and is considered a crucial marker for understanding how the brain quickly identifies and processes facial features ([23]). The N170, peaking around 170 ms, is particularly sensitive to faces and reflects higher-level processing, such as structural encoding and facial feature detection ([40]). The N170 is typically larger for faces compared to other objects, highlighting its specialized role in face perception ([40]). Strong evidence suggests that the N170 is generated by neural sources in the fusiform gyrus ([17]; [19]), which is, as mentioned above, a key region for face processing. Finally, the P300, occurring around 300 ms, is linked to the cognitive evaluation of the stimulus, including categorization and emotional processing ([38]).

### 1.3. Face Components of Emotional Expression

Non-verbal communication, particularly through facial emotional expressions, is essential for conveying emotional states, enabling individuals to quickly and effectively express and interpret feelings in social interactions ([16]). These expressions foster empathy, strengthen interpersonal connections, and guide appropriate responses in complex social contexts.

Facial features, especially the eyes and mouth ([9]; [21]), are key in conveying and distinguishing between emotional expressions. The eyes are highly expressive, with changes in gaze direction ([4]), eyebrow movement, and eyelid position offering important cues for emotions such as surprise ([33]), fear ([1]), and sadness. The mouth also plays a central role in emotional expression, with changes in lip shape and movement signaling emotions such as happiness (e.g., smiling), anger (e.g., pursed lips), or disgust (e.g., curled lips). The interaction between eye and mouth movements allows for nuanced communication of complex emotional states, facilitating rapid interpretation of feelings and intentions.

Studies suggest that the eyes and mouth contribute differently depending on the emotion being expressed. For example, identifying neutral or happy expressions often relies more on the mouth region, while emotions such as fear, surprise, and anger are more dependent on the eyes ([20]; [28]). In clinical populations, such as individuals with autism or prosopagnosia, difficulties in processing the eye region have been associated with challenges in distinguishing between emotional expressions ([10]), underscoring the importance of the eyes in emotion recognition. However, other studies have proposed that the mouth may play an equally important or even more significant role in perceiving emotional expressions ([5]), suggesting that the relative importance of facial features may depend on the specific emotional context. However, the longstanding belief that humans universally express and recognize specific emotions through fixed facial configurations, such as smiles for happiness or scowls for anger, lacks robust scientific support. Research shows that facial expressions of emotion are highly variable and context-dependent, shaped by cultural, situational, and individual factors ([2]), which makes studying how emotional facial expressions affect Event-Related Potentials (ERPs), discussed in the next paragraph, more challenging.

### 1.4. Effect of Emotional Expressions on Event-Related Potentials

The question of whether emotional facial expressions influence ERP components, specifically the P100 and N170, has been the subject of extensive research, with varying results. A comprehensive review by Schlinder and Bublatzky ([43]) examined ERP responses to emotional and neutral faces under various conditions of attentional control. This review included 102 studies published between 2009 and 2019. Based on this large body of research, the authors concluded that the P100 reflects early, resource-independent visual identification of facial stimuli and does not reliably show emotional effects across tasks. In contrast, the N170, associated with early configural processing of faces, was consistently modulated by emotional expressions, particularly in passive viewing paradigms. This suggests that while the P100 captures basic visual processing, the N170 is sensitive to both the structural and emotional features of faces, representing an early stage of emotional processing. Also, an earlier meta-study involving 57 experiments focused on the N170 ([24]) shows that the N170 response is particularly sensitive to emotional facial expressions such as anger, fear, and happiness, indicating that it does more than simply detect changes in faces; it also helps process socially important expressions. The authors conclude that it challenges the idea that the brain processes facial identity and facial expressions separately (dual models) but instead supports the view that these processes are interconnected and rely on shared mechanisms (integrated models).

### 1.5. The Research Question

Despite most studies finding that emotional facial expressions elicit a significantly larger N170 amplitude compared to neutral faces, the results across studies are highly variable. For example, in the meta-analysis by Hinojosa and colleagues ([24]), about 30 out of 57 studies reported a larger N170 amplitude for emotional faces, while around 10 studies interestingly found the opposite—a larger N170 for neutral faces (see Figure 2 in their paper). While differences in study designs and face stimuli likely account for some of this variability, we wondered if a key variable might have been overlooked in many of these studies.

Our idea for this study emerged after examining the sample stimuli in Figure 1 of Blau et al.’s study ([6]). In their “fear condition”, the faces clearly show exposed teeth, whereas the neutral condition does not. While these are just examples and may not apply to all images used, it is well-documented that facial expressions such as smiling ([44]) or certain negative emotions ([31]) are more likely to involve open mouths or exposed teeth. Additionally, a 2012 study even suggests that the emotional-face-in-a-crowd effect—where angry faces are quickly detected in a crowd—might be driven by the visibility of teeth rather than the emotional expression itself ([25]).

This led us to hypothesize that the visibility of teeth might be a confounding variable in some emotional face ERP studies. To test this, we designed a 3 × 2 study to control for this variable and investigate whether the visibility of teeth, rather than the emotional expression, could drive changes in N170 amplitude. We hypothesized that exposed teeth, regardless of the emotional expression, would result in a larger N170 amplitude.

## 2. Methods and Materials

### 2.1. Subjects

A total of 42 participants took part in this study, divided between two versions of the experiment. The first version included 19 participants, the majority of whom were female (13) and Hispanic (15), with an average age of 26 ± 11 years. The second version included 23 participants, with a similar demographic distribution: 19 were female, 15 were Hispanic, and the average age was 25 ± 11 years.

All participants were psychology majors at California State University, San Bernardino (CSUSB), recruited via an online platform. Participation was voluntary, with participants either receiving class credit as compensation or choosing to participate without compensation.

Prior to the experiment, participants were given a brief overview of the study’s purpose, which focused on using electroencephalogram (EEG) to explore the neural mechanisms underlying facial recognition. Informed consent was obtained from all participants. Each participant completed only one version of the experiment to avoid redundancy.

The study was approved by the Institutional Review Board (IRB) at CSUSB.

### 2.2. Stimuli

For each version of the experiment, 288 unique images of faces were used, balanced across gender, race/ethnicity (Black, Caucasian, Asian, Hispanic), emotional expressions (“Fearful”, “Happy”, “Neutral”), and mouth display conditions (“closed mouth” and “open mouth with exposed teeth”). The faces were selected from the RADIATE database ([11]), an open-access resource featuring racially and ethnically diverse models displaying various emotional expressions with both open and closed mouths.

A key selection criterion for the “open mouth” condition was that the images clearly displayed exposed teeth. The stimuli used for both versions were identical, with two exceptions:In the second version, faces were cropped using Photoshop to exclude non-facial features such as hair, presenting only the face. This approach replicated the stimuli format used by [6] ([6]).The background in the second version was set to 50% gray instead of white.

Example stimuli for both versions are shown in Figure 1.

### 2.3. Experimental Procedure and EEG Equipment

Participants wore a 64-channel EEG cap (BrainVision, Garner, NA, USA) configured to use 32 active channels, sampled at 500 Hz. The reference electrode was placed at the FCz position, following the 10–20 EEG system. Before starting the recording, all electrodes were connected to a BrainVision actiCHamp active channel amplifier and checked to ensure proper conductivity, with impedances maintained below 5 kΩ.

The experiment was conducted in a dimly lit, quiet room. Participants were seated comfortably, facing a 19-inch Dell monitor placed 50 cm from their eyes. The experimental paradigm was programmed using Experiment Builder (SR Research).

Two-hundred-and-eighty-eight face images (see “Stimuli”) were presented sequentially at the center of the screen, each subtending a visual angle of 17° × 23°. Image order was randomized to mitigate potential biases. To maintain attention, participants were instructed to identify the gender of each face using a button box (see Figure 1B). Each face remained on the screen for a minimum of 1 s and disappeared as soon as a response was recorded. If no response was provided within 4 s, the face disappeared automatically. Between trials, a fixation cross was displayed for 1 s at the center of the screen. Participants were encouraged to respond as quickly and accurately as possible.

### 2.4. Analysis

#### 2.4.1. Data Segmentation and Preprocessing

The EEG data were analyzed offline using the FieldTrip Matlab (R2018b) software toolbox ([34]). Noisy trials were identified and removed using the FieldTrip data browser function. Timestamps from the photodiode input were utilized to segment the data, obtained during each session, into 288 trials, each lasting 2 s. To mitigate edge effects during preprocessing, these segments spanned from 0.5 s before stimulus onset to 1.5 s after onset. The raw data were then filtered (3–45 Hz) and demeaned. A high-pass filter at 3 Hz effectively removed slow drifts observed in the EEG signal ([36]), supplemented by DFT notch filters at 60 and 120 Hz.

#### 2.4.2. Electrode Selection and Grand Average ERP Waveforms

ERP waveforms and their components were calculated by averaging data from a cluster of occipitotemporal electrodes (see Figure 1C), following the 10–20 system and closely aligned with established practices in the field ([18]).

For the 3 × 2 factorial design, data were averaged separately for each participant across the three emotional expression conditions and the two mouth display conditions. Grand average ERP waveforms (Figure 2 and Figure 3) were subsequently computed by averaging across participants for each experimental version (Figure 2A–D) or by pooling data from both versions (Figure 2E,F, Table 1 and Table 2, and Figure 3). To ensure smooth waveforms, a 5-point moving average with a 0.010-s window was applied.

#### 2.4.3. Statistical Analysis

To evaluate statistical differences in ERP waveform amplitudes across experimental conditions, we employed a “running *p*-value” approach using paired *t*-tests. These tests were conducted with sample sizes of n = 19 (Figure 2A,B), n = 23 (Figure 2C,D), and n = 42 (Figure 2E,F), corresponding to the respective participant groups.

The time bins compared were derived using a moving mean calculated over a sliding window of 10 samples (equivalent to 20 ms), resulting in 162 consecutive 2-ms epochs spanning the interval from −25 ms to 300 ms relative to stimulus onset. For each epoch, significant differences between conditions were marked beneath the waveforms using vertical lines. These markers were color-coded to indicate the level of significance: yellow for *p* < 0.05, orange for *p* < 0.01, and red for *p* < 0.001.

The “running *p*-value” analysis serves primarily as a visual tool to provide readers with an intuitive overview of the temporal dynamics of the observed differences between experimental conditions. However, we acknowledge that this approach involves multiple comparisons and may result in significant differences occurring by chance alone. To address this limitation, our primary statistical analyses focused on the peak values of the P100 and N170 components, addressed below.

The study employed a 3 × 2 factorial design to examine the interactions between the independent variables concerning P100 and N170 amplitudes. A two-way repeated measures ANOVA (rANOVA) was performed using the MATLAB function ranova.m (results shown in Table 1 and Table 2). For this analysis, the dependent variables were the P100 and N170 amplitudes for each participant and each experimental condition. P100 amplitude values were calculated as the maximum voltage within the 80–120 ms time window following stimulus onset, while N170 amplitude values were derived as the minimum voltage within the 100–200 ms time window.

#### 2.4.4. Non-Parametric Analysis

To evaluate the statistical significance of the latency differences between the “Fearful” condition and the “Happy” and “Neutral” conditions, we employed a non-parametric approach with 10,000 iterations implemented in MATLAB. The following steps were performed during each iteration:The 288 trials from each participant were randomly divided into three pseudo-conditions—“Fearful”, “Happy”, and “Neutral” (96 trials each). Using the same approach applied to the actual data, a grand average ERP waveform was generated for each pseudo-condition.For each waveform, the N170 peak latency was identified as the time point corresponding to the minimum voltage (in microvolts) within the 100–200 ms time window following stimulus onset.The segment from the N170 peak latency to 200 ms post-stimulus was extracted for each waveform.These segments were further refined to retain only portions with amplitudes between −3 and 1 microvolts, approximately corresponding to the dashed rectangle in Figure 3.The refined segments for each pseudo-condition were averaged, resulting in a single latency value for each pseudo-waveform.The time delay was then calculated by subtracting the latency value for the “pseudo-fearful” condition from the average latency of the “pseudo-happy” and “pseudo-neutral” conditions.

The results of the 10,000 iterations are shown as a histogram in the inset of Figure 3, with the red vertical line indicating the observed latency difference in the actual data.

## 3. Results

Our study was primarily inspired by a publication by Blau et al. ([6]), which has been widely cited as evidence that the N170 response is strongly modulated by emotional expression, with a larger N170 response for emotional (e.g., fearful) faces than for neutral expressions. However, their study did not control for the presence of exposed teeth, which is more likely to appear in the “Fearful” condition and not in the “Neutral” one, potentially introducing a confounding variable. To examine whether exposed teeth might account for the N170 modulation observed, we designed an experiment with two versions. In each, participants categorized faces by gender using a button box. In the first version, full faces were presented, while in the second, faces were cropped (see Methods). Across both versions, participants viewed happy, fearful, and neutral faces, each balanced for mouth position (closed, or open with exposed teeth), and randomized across participants.

We analyzed the results for the independent variables of emotion (Figure 2, left panels) and mouth position (Figure 2, right panels) separately for each version (Version 1: panels A, B; and Version 2: panels C, D) and for both versions combined (panels E and F). Differences in the emotion variable were most pronounced between “Fearful” compared with “Happy” and “Neutral” conditions, with only minor differences between the “Happy” and “Neutral” conditions. Visual inspection of the N170 component suggested that these differences were due to latency shifts rather than amplitude changes, prompting further analysis (see below). For the mouth position variable, we observed a larger N170 amplitude for open-mouth (exposed teeth) faces, regardless of emotional expression. When data from both versions were combined, this difference reached statistical significance for the time epoch corresponding to the ERP N170 peak.

**Figure 2 behavsci-15-00163-f002:**
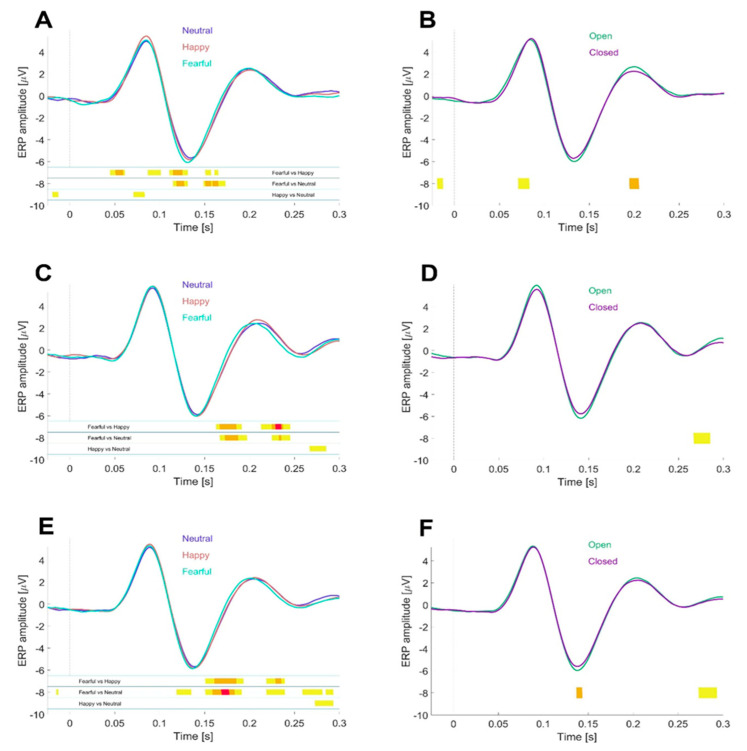
ERP Waveforms Across Experimental Conditions. (**A**): Grand-mean ERP waveforms averaged across 19 participants (Version 1) and a cluster of occipitotemporal channels (see Methods and Figure 1C for electrode selection). Waveforms are split by emotional expression condition (“Neutral” in purple, “Happy” in red, and “Fearful” in cyan), with 96 trials per condition per participant. (**B**): Same data as (**A**), but split by mouth display condition (“closed mouth” in purple, “open mouth with exposed teeth” in cyan), with 144 trials per condition. (**C**,**D**): Equivalent data to (**A**,**B**), respectively, but from Version 2 of the experiment, with 23 participants. (**E**,**F**): Combined results across both experimental versions, pooling data from all 42 participants. (**E**) Aggregate results from (**A**,**C**), while (**F**) combines (**B**,**D**). Statistical testing of amplitude differences between experimental conditions was conducted using sliding paired *t*-tests (see Methods). Time intervals with statistically significant differences are indicated by vertical lines beneath the waveforms, color-coded as follows: yellow (*p* < 0.05), orange (*p* < 0.01), and red (*p* < 0.001). For the independent variable “emotion” with three conditions (see **A**,**C**,**E**), pairwise comparisons were performed between conditions, resulting in three comparisons. The dotted vertical line in each panel indicates the timing of stimulus onset.

To assess whether the amplitudes of the P100 and N170 components were influenced by emotional expression or the presence of exposed teeth, we conducted a repeated measures ANOVA on the combined dataset (see Table 1 and Table 2, and Methods for amplitude determination). Results showed that the N170 amplitude was not significantly modulated by emotional expression (*p* = n.s.) but, on the other hand, significantly (*p* < 0.01) affected by the presence of exposed teeth (open mouth condition), with no interaction between the two independent variables, emotional expression and mouth display condition (*p* = n.s.). In contrast, the ANOVA for the P100 amplitude revealed a significant interaction between those two independent variables with *p* < 0.05.

**Table 1 behavsci-15-00163-t001:** ANOVA results for P100 amplitudes.

Effect	df	SS	MS	F	*p*	η^2^
Participant	41	1118.631	27.284			
Emotion (fearful, happy, neutral)	2	0.8	0.4	0.445	0.6421	0.0006
Participant (Emotion)	82	73.676	0.898			
Mouth (open, closed)	1	0.655	0.655	0.699	0.408	0.0005
Participant (Mouth)	41	38.447	0.938			
Emotion: Mouth	2	5.995	2.997	3.213	0.0454	0.0046
Participant (Emotion: Mouth)	82	76.496	0.933			

The P100 amplitudes were calculated for each condition and participant (see Methods). Differences in amplitudes across the conditions were analyzed using a two-way repeated measures analysis of variance (rANOVA). The last column lists the effect sizes, as measured by partial eta squared (η^2^).

**Table 2 behavsci-15-00163-t002:** ANOVA results for N170 amplitudes.

Effect	df	SS	MS	F	*p*	η^2^
Participant	41	2530.883	61.729			
Emotion (fearful, happy, neutral)	2	4.322	2.161	2.934	0.0588	0.0016
Participant (Emotion)	82	60.409	0.737			
Mouth (open, closed)	1	6.93	6.93	10.032	0.0029	0.0026
Participant (Mouth)	41	28.322	0.691			
Emotion: Mouth	2	0.173	0.087	0.159	0.8534	0.0001
Participant (Emotion: Mouth)	82	44.777	0.546			

Similar to Table 1 but focusing on N170 amplitudes.

To determine whether the latency shift between the “Fearful” and “Happy”/”Neutral” conditions, as observed through visual inspection in Figure 2E (starting about 150 ms post-stimulus), is significant, we conducted a Monte Carlo simulation (see Methods and Figure 3). This analysis revealed that only 67 and 5 out of 10,000 iterations produced an average time delay for the “Happy” and “Neutral” conditions vs. “Fearful”, respectively, that was equal to or larger than the observed delay of 2 ms (see histogram, inset Figure 3).

**Figure 3 behavsci-15-00163-f003:**
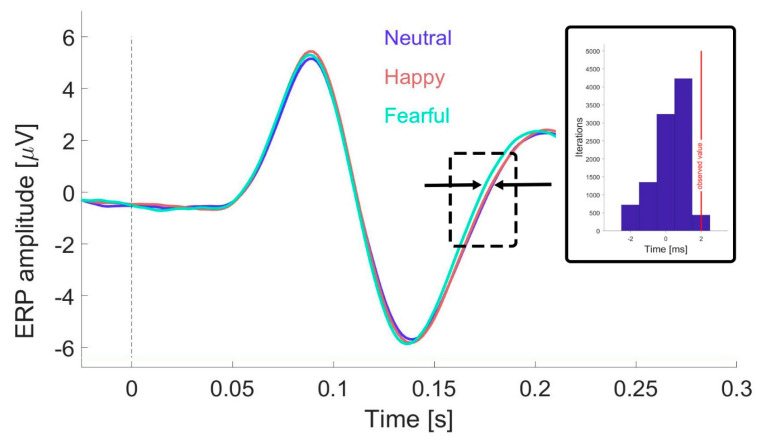
Delay between ERP waveforms across emotional expression conditions. To explore the delay observed between the “Fearful” condition and the other two conditions (“Happy” and “Neutral”) within the time window highlighted by the dotted rectangle (copied from Figure 2E), Monte Carlo simulations with 10,000 iterations were performed (see Methods for details). The inset displays the results of these simulations as a histogram, with the observed delay of 2 ms indicated by a red vertical line.

## 4. Discussion

### 4.1. Resolving Inconsistencies in ERP Studies of Emotional Faces

The primary aim of this study was not merely to contribute another demonstration of how emotional expressions modulate early ERP components, such as the N170, but also to address the inconsistencies reported in the literature. Previous studies have found variable effects of emotional facial expressions on N170 amplitude, with some reporting larger amplitudes for emotional faces compared to neutral ones, while others have observed no effect or even the opposite pattern ([24]). These discrepancies may stem from methodological differences, such as variations in study design, stimuli, or analysis approaches ([24]; [43]). Consistent with these inconsistencies, our previous work also found no significant differences in P100 or N170 amplitudes between ‘happy’ and ‘angry’ faces, further underscoring the variability in the field ([7]).

One plausible explanation for these inconsistencies is that neutral faces are more likely to have closed mouths, and consequently lack visible teeth, while emotional expressions often feature exposed teeth.

Our findings suggest that the presence of teeth, rather than the emotional expression itself, may confound ERP responses. By carefully controlling this variable, we demonstrated that when mouth positions (closed vs. open with teeth exposed) are balanced across all emotional expressions, including neutral ones, the N170 is not modulated by emotional expression. Instead, the N170 amplitude was significantly larger for stimuli with exposed teeth, irrespective of emotional expression. Furthermore, our results confirm that mouth display position and emotional expression are independent variables that do not interact to influence ERP responses. By systematically controlling for this variable, we demonstrated that the N170 amplitude is not directly modulated by emotional expression but is significantly influenced by the presence of exposed teeth, irrespective of the emotional expression being displayed ([25]).

It is worth noting that the effect sizes, as measured by partial eta squared (η^2^) for N170 amplitudes, were very small: 0.0026 for mouth position, 0.0016 for emotional expression, and 0.0001 for the interaction between these variables. Despite these small effect sizes, the difference in N170 amplitude for mouth position (closed mouth vs. exposed teeth) was highly significant (*p* < 0.01). This paradox of small effect sizes alongside significant results raises the possibility that the subtle influence of mouth position on N170 amplitude may have been overlooked in previous studies. This could potentially account for some of the inconsistencies in findings across studies, particularly in research that did not systematically control for the presence of visible teeth ([6]; [43]). These results underscore the importance of examining even small but statistically robust effects to refine our understanding of face-processing mechanisms.

### 4.2. The Influence of Teeth Visibility on P100 and N170 Amplitude

Our conclusion aligns with findings from prior studies. For instance, research investigating the effects of mouth opening in emotional faces ([29]) showed that an open mouth—independent of emotional expression—enhanced subjective emotional intensity and early attentional capture. This further supports the notion that visual features such as teeth critically modulate neural responses such as N170.

Building on this, Crager et al. ([12]) used a design isolating mouth configuration (grimace, smile, open mouth) and teeth visibility with stimuli showing only the lower half of the face. Their findings corroborate our results, revealing that visible teeth influence early ERP components, including the N170, and increase subjective arousal ratings. These results highlight the salience of teeth as a visual feature in facial processing and underscore the importance of systematically controlling these elements in future research.

The observed interaction effect between emotional expression and mouth position on P100 amplitude, albeit with a *p*-value just below 0.05, suggests that the early stages of facial processing may be influenced by the combination of emotional content and specific facial features. This interaction indicates that the P100 component might reflect an early integration of basic visual features and emotional cues, potentially influencing subsequent stages of face processing. While the effect size is modest, this finding underscores the need to further investigate the role of the P100 in processing emotional expressions, particularly in contexts where visual salience is modulated by facial features such as teeth visibility.

### 4.3. Holistic Versus Feature-Driven Processing: Discrepancies Across Methodologies

Interestingly, a recent study ([35]) using facial stimuli without visible teeth found that fearful faces elicited increased N170 amplitudes. The researchers used a facial decoding technique with a large sample and extensive stimuli, concluding that these amplitude differences reflected holistic face processing rather than modulation by specific facial regions. This discrepancy with our findings may stem from methodological differences: while they employed single-trial analyses with randomly distributed facial information across spatial frequency scales, our approach relied on traditional ERP averaging with complete facial stimuli. These contrasting methodologies highlight the need to consider experimental approaches when interpreting ERP results.

### 4.4. Role of Object-Sensitive Neurons in N170 Modulation?

A potential explanation for the increased N170 response to visible teeth involves the additional recruitment of object-sensitive neurons during the N170 time window. This hypothesis has been proposed ([41]) to account for the heightened N170 amplitude observed for inverted faces and may also explain the increased N170 seen in faces with long versus short hair ([48]). This interpretation aligns with the idea that the presence of prominent visual features, such as teeth, activates neural mechanisms typically associated with object recognition.

### 4.5. Temporal Shifts in Processing Emotional Expressions

A second conclusion from our study, which controlled for mouth position, is that the differences observed between the emotional conditions—particularly between “Fearful” and the other two conditions, “Neutral” and “Happy”—can be attributed to a temporal shift rather than modulation of ERP components. Specifically, during the time window including and immediately following the N170, fearful faces appeared to be processed approximately 2 ms faster than the other conditions. Temporal shifts, including latency delays, have similarly been observed in response to inverted faces ([15]) or unexpected face stimuli ([8]), reflecting the brain’s adaptation to salient or atypical visual inputs.

This finding aligns with evolutionary perspectives, where rapidly processing potentially threatening stimuli could significantly enhance survival in dangerous situations. Supporting this, direct evidence for a subcortical pathway to the amygdala that facilitates the rapid detection of fearful faces—even when rendered invisible through backward masking ([50])—corroborates our observation of faster processing of fearful faces. This suggests that subcortical mechanisms prioritize detecting potential threats and optimizing survival outcomes.

### 4.6. The Role of Teeth Visibility and Emotional Intensity in Modulating the N170: Insights, Limitations, and Future Directions

Our study investigated structural factors, such as teeth visibility, and their influence on N170 amplitude. However, we acknowledge the well-documented role of emotional intensity in modulating the N170. For instance, a meta-analysis ([24]) demonstrated that emotional intensity reliably enhances N170 amplitude, particularly for highly arousing expressions such as fear or anger. This aligns with findings from studies using well-validated datasets, such as J3DFD ([32]), which reveal that emotional intensity interacts with early face-processing stages. However, these studies often fail to control for structural features such as teeth visibility, potentially conflating the effects of emotional intensity with visual salience.

Our findings contribute to this body of work by demonstrating that teeth visibility independently modulates N170 amplitude. However, we did not directly manipulate emotional intensity, which represents a limitation of our study. This oversight is particularly relevant since expressions with open mouths may systematically convey greater emotional intensity than their closed-mouth counterparts. Future research should adopt experimental designs that simultaneously control for both teeth visibility and emotional intensity to better understand their interaction and relative contributions to ERP responses.

Disentangling the effects of teeth visibility and emotional intensity is particularly challenging because certain expressions (e.g., happiness and fear) inherently involve the exposure of teeth. The use of AI-generated faces, designed to explicitly control these variables, could help clarify, rather than confound, whether open mouths and exposed teeth act as mediators rather than confounds. Such approaches could further elucidate the specific contributions of emotional intensity and structural features to ERP responses.

## Figures and Tables

**Figure 1 behavsci-15-00163-f001:**
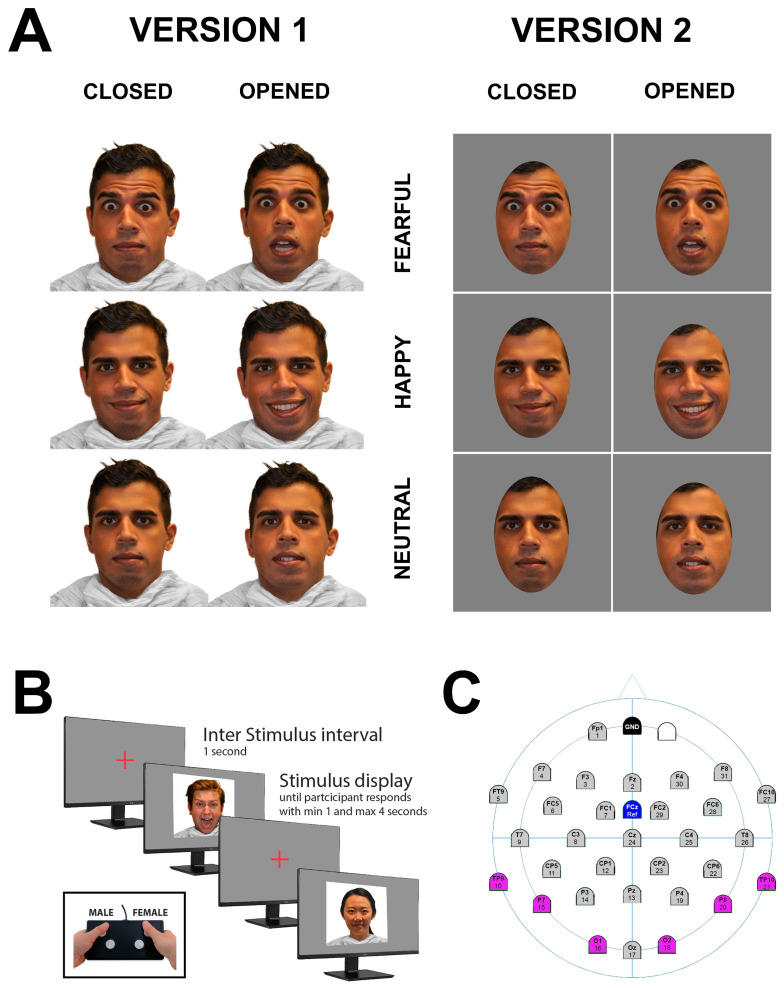
Stimulus Selection and Experimental Setup. (**A left**): Example stimuli used in Version 1, illustrating the experimental conditions based on the independent variables: emotional expression (“Fearful”, “Happy”, and “Neutral”) and mouth display condition (“closed mouth” and “open mouth with exposed teeth”). (**A right**): Example stimuli used in Version 2, which are the same as those in Version 1 but cropped and presented with a different background (see Methods for details). (**B**): Stimulus presentation and response task for both versions. A total of 288 stimuli are presented in random order. Participants use a button box to indicate whether the face is male or female. Faces are displayed for a minimum of 1 s and remain on screen until the participant responds or for a maximum of 4 s if no response is recorded. The intertrial interval is 1 s, during which a fixation cross appears at the center of the monitor. (**C**): Topographic map of sensor locations. This panel shows the topographic map of sensor locations used in the study. Results (Figure 2 and Figure 3, and Table 1 and Table 2) were derived from data averaged across the sensors located above the occipitotemporal scalp regions (highlighted in purple).

## Data Availability

The authors will make the data and code used for the analysis in this study available upon reasonable request. Interested researchers are encouraged to contact the corresponding author for access to these materials, which will be provided to promote transparency and facilitate further exploration of the findings.

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
