# Peer review of "Effects of Closed Mouth vs. Exposed Teeth on Facial Expression Processing: An ERP Study"

_behavsci, 2025, doi:10.3390/bs15020163_

Round 1

Reviewer 1 Report

Comments and Suggestions for Authors

The authors have addressed the debate among researchers on whether the N170 encodes emotional facial information, with both the introduction and discussion sections being well-written. I have a few minor questions:

The first issue I'm concerned about is the sample size. The authors recruited 19 and 23 participants in Study 1 and Study 2, respectively, which is a relatively small sample size.

I recommend using currently mainstream statistical analysis methods to determine the required number of participants for the experiment.

The electrode selection covers a relatively broad area, including a total of 11 electrodes, which is different from most N170-related studies we usually see. It also differs somewhat from the two references cited by the authors, 43 and 44. My question of interest is whether all these electrodes are necessary for N170 analysis, such as whether FT9 and FT10 must be included in the analysis. Additionally, although the authors pointed out that the references show the location of the electrodes in Figure 1C, due to resolution issues, we still cannot clearly see which electrodes were selected. I suggest the authors supplement this information in the main text or the figure caption.

The presentation of the statistical analysis results is not very clear. I find it difficult to associate the time segments of interest in Figure 2 with whether there are differences in the results, partly due to resolution issues and partly because the authors have not clearly marked the time ranges of interest for N170 or P1. Additionally, I need to confirm that the running p-value statistical analysis in this study only serves the purpose of presenting the results, and the main statistical analysis results are obtained using ANOVA analysis?

Author Response

We would like to express our sincere gratitude to the reviewer for valuable feedback and suggestions, which have significantly strengthened our manuscript. To enhance the readability of this document, we have used blue font for the reviewers' comments and purple font for our responses.

REVIEWER 1: The authors have addressed the debate among researchers on whether the N170 encodes emotional facial information, with both the introduction and discussion sections being well-written. I have a few minor questions:

The first issue I'm concerned about is the sample size. The authors recruited 19 and 23 participants in Study 1 and Study 2, respectively, which is a relatively small sample size.

I recommend using currently mainstream statistical analysis methods to determine the required number of participants for the experiment.

ANSWER: We appreciate the reviewer’s feedback regarding our sample sizes. It is important to note that EEG studies with approximately 20 participants per condition are common in the field, as the high signal-to-noise ratio achieved through trial averaging makes this a standard practice. Meta-analyses, such as Schlinder and Bublatzky (2020), report that sample sizes of this magnitude are typical for ERP studies investigating face processing.

More importantly, our key findings are derived from the pooled dataset of 42 participants, combining data from Studies 1 and 2. This pooled analysis not only enhances statistical power but also exceeds the sample sizes in many comparable EEG studies. By combining the datasets, we ensure that the conclusions are based on a robust and sufficiently large sample.

Additionally, our analyses are grounded in effect sizes and statistical power calculations consistent with established norms in ERP research. The reported effect sizes in our manuscript (added to Table 1, in response to a request of Reviewer 2) indicate that the sample size was sufficient to detect significant differences for our primary effects of interest. Taken together, the sample size and analytical methods align with accepted standards in the field and provide a reliable foundation for our conclusions.

REVIEWER 1: The electrode selection covers a relatively broad area, including a total of 11 electrodes, which is different from most N170-related studies we usually see. It also differs somewhat from the two references cited by the authors, 43 and 44. My question of interest is whether all these electrodes are necessary for N170 analysis, such as whether FT9 and FT10 must be included in the analysis. Additionally, although the authors pointed out that the references show the location of the electrodes in Figure 1C, due to resolution issues, we still cannot clearly see which electrodes were selected. I suggest the authors supplement this information in the main text or the figure caption.

ANSWER: We appreciate the reviewer’s suggestion regarding the selection of electrodes for N170 analysis. Following this advice, we reduced the number of electrodes to include only P7, P8, O1, O2, TP9, and TP10, which aligns more closely with common practices in N170-related studies. We reanalyzed all the data using this refined electrode cluster, resulting in slightly different results, updated graphs (Figures 2 and 3), and modified numbers in Table 1.

While the specific values changed due to the adjustment, these changes are minimal, confirming the robustness of our findings. The updated electrode selection and results have been detailed in the revised text.

REVIEWER 1: The presentation of the statistical analysis results is not very clear. I find it difficult to associate the time segments of interest in Figure 2 with whether there are differences in the results, partly due to resolution issues and partly because the authors have not clearly marked the time ranges of interest for N170 or P1. Additionally, I need to confirm that the running p-value statistical analysis in this study only serves the purpose of presenting the results, and the main statistical analysis results are obtained using ANOVA analysis?

ANSWER: We thank the reviewer for raising this important point. To clarify, the purpose of the "running p-value statistical analysis," visualized beneath the ERP waveforms in each panel of Figure 2, is to provide readers with a quick and intuitive visualization of the observed differences between experimental conditions. As the reviewer correctly noted, this approach, due to the multiple comparisons involved, can result in significant differences arising by chance alone.

To address this limitation, our primary statistical analyses focused on the P100 and N170 peak values, which were identified as outlined in the Methods section. The ANOVA results reported in the manuscript are based on these peak values, ensuring that our conclusions are drawn from statistically robust analyses. We have clarified this in the revised text for transparency by inserting the following text in the “Statistical analysis” section:

“The "running p-value" analysis serves primarily as a visual tool to provide readers with an intuitive overview of the temporal dynamics of the observed differences between experimental conditions. However, we acknowledge that this approach involves multiple comparisons and may result in significant differences occurring by chance alone. To address this limitation, our primary statistical analyses focused on the peak values of the P100 and N170 components.”

Reviewer 2 Report

Comments and Suggestions for Authors

This ERP study investigated whether the visibility of teeth, rather than emotional expression itself, might explain previously reported differences in N170 amplitudes between emotional and neutral faces. The key finding was that N170 amplitude was significantly larger for faces showing teeth regardless of emotional expression, suggesting that many prior studies examining emotional face processing may have been confounded by not controlling for teeth visibility. The study was expertly designed and conducted. I commend the author's efforts to replicate the findings using two versions of faces (full and cropped) and with a good sample size of 42 participants. Further, they present careful statistical analysis, including both parametric and non-parametric methods. This study potentially reveals a crucial methodological consideration that has implications for both interpreting past research and designing future studies of emotional face processing and, thus, is of great relevance to the field. However, I am not entirely convinced that the reported effect is merely due to the presence of the teeth and independent of the emotion itself. As described below, I worry that the emotional intensities are not matched across conditions, potentially introducing an uncontrolled confounding variable that can explain the observed effects. Therefore, my recommendation is that the authors address this issue with an independent rating task to verify that emotional intensity across conditions is matched.  This will allow them to make a strong claim about the role of teeth visibility (or mouth opening) and allay concerns about the confounding emotional intensity variable. 

Core Issue:

Faces with open vs. closed mouths may differ systematically in the intensity of emotional expression. For example, fearful faces with open mouths may express more 'intense' fear than those with closed mouths. Further, this may also hold for other emotions, such as happiness. Indeed, I feel that in Figure 1 A, the intensity of fear and happiness in the open-mouthed models is higher than the corresponding close-mouthed counterparts. Is this true across all stimulus images? If so, this is a major impediment to the conclusion that teeth visibility rather than emotional expression drives N170. In other words, if open-mouthed expressions were systematically more intense than close-mouthed expressions, the "teeth effect" could actually be an intensity effect within each emotion category. How can this confound be addressed?

Ideally, my approach would have been to do a pre-study stimulus validation, where a group of independent raters scores the intensity of emotions for open and closed versions of emotions. Only faces with similar intensity ratings for these two versions should be selected. 

If the above was not done pre-study, I would recommend using posthoc statistical methods to control for this confound. Again, have a group of independent raters score the emotional intensities and compare open vs. closed mouth conditions. If there is no statistical difference, the confound is addressed. Otherwise, the variance explained by emotional intensity can be regressed out, and the authors can demonstrate that the presence or absence of teeth explains additional variance in a model fitted on the residuals. 

Minor Issues:

1. Please include a partial eta-squared for the open vs. closed main effect. This will help gauge if the effect size is meaningful. 

2. I see a significant P100 interaction effect described in the results. However, the manuscript does not discuss the potential implications of this finding.  Please include a few lines about the P100 effect and what it may mean. This will also give closure to the reader given the succinct exposition on the role of P100 and N170 components in emotional expression processing in section 1.4. 

Author Response

We would like to express our sincere gratitude to the reviewer for valuable feedback and suggestions, which have significantly strengthened our manuscript. To enhance the readability of this document, we have used blue font for the reviewers' comments and purple font for our responses.

REVIEWER 2: Core Issue: Faces with open vs. closed mouths may differ systematically in the intensity of emotional expression. For example, fearful faces with open mouths may express more 'intense' fear than those with closed mouths. Further, this may also hold for other emotions, such as happiness. Indeed, I feel that in Figure 1 A, the intensity of fear and happiness in the open-mouthed models is higher than the corresponding close-mouthed counterparts. Is this true across all stimulus images? If so, this is a major impediment to the conclusion that teeth visibility rather than emotional expression drives N170. In other words, if open-mouthed expressions were systematically more intense than close-mouthed expressions, the "teeth effect" could actually be an intensity effect within each emotion category. How can this confound be addressed?

Ideally, my approach would have been to do a pre-study stimulus validation, where a group of independent raters scores the intensity of emotions for open and closed versions of emotions. Only faces with similar intensity ratings for these two versions should be selected. 

If the above was not done pre-study, I would recommend using posthoc statistical methods to control for this confound. Again, have a group of independent raters score the emotional intensities and compare open vs. closed mouth conditions. If there is no statistical difference, the confound is addressed. Otherwise, the variance explained by emotional intensity can be regressed out, and the authors can demonstrate that the presence or absence of teeth explains additional variance in a model fitted on the residuals. 

ANSWER: We appreciate the reviewer’s thoughtful comments regarding the potential confound of emotional intensity between open- and closed-mouth conditions in our study. While we acknowledge the validity of this concern, addressing it through additional data collection poses significant logistical challenges. Specifically, obtaining Institutional Review Board (IRB) approval for a new behavioral study, recruiting participants, and conducting data collection would require several weeks, which is problematic given the deadline for the special issue to which our manuscript was submitted.

Nonetheless, we recognize the importance of addressing this issue and offer several points of consideration based on prior research. First, Shuang et al. (2023) investigated the intensity effects of the NimStim facial expression set (Tottenham et al., 2009), and found that while the intensity of expressions significantly influenced later ERP components such as VPP and LPP, it did not affect the N170. This result suggests that differences in emotional intensity related to mouth opening or closing are unlikely to explain the modulation of N170 observed in our study. Our findings align with this, as the N170 was significantly modulated by teeth visibility, with no interaction between emotional expression and mouth condition. This further supports the conclusion that the N170 response in our study reflects structural processing differences rather than emotional intensity.

Second, Langeslag et al. (2018) examined how mouth opening influences subjective emotional experience and attentional capture, as measured by the early posterior negativity (EPN). They demonstrated that open-mouth expressions elicited stronger valence and arousal responses in observers, as well as enhanced EPN amplitudes, particularly for emotional faces. However, like Shuang et al., they found these effects were more pronounced in later ERP components. Their findings highlight that the effects of mouth opening on early attentional capture are distinct from those related to structural encoding, as reflected by the N170.

Taken together, these studies support the interpretation that the observed N170 effects in our study are primarily driven by structural processing differences associated with teeth visibility, rather than differences in emotional intensity. While conducting a behavioral validation study could provide additional insights, evidence from Shuang et al. (2023) and Langeslag et al. (2018) strengthens the argument that the modulation of N170 in our study is unlikely to be confounded by expression intensity. To further address this issue, we took the following action: we acknowledge the possibility of an alternative explanation for the results in the section “Shortcomings, Alternative Explanations, and Future Directions” which replaces the original “Shortcomings and Future Direction section). In addition, we also revised the abstract to briefly acknowledge the possibility of an alternative explanation for the results. We hope that these steps clarity while aligning with the constraints of the current timeline.

TEXT REPLACING THE ORIGINAL “SHORTCOMINGS AND FUTURE DIRECTION” section

Shortcomings, Alternative Explanations, and Future Directions

While the findings of this study provide new insights into the influence of teeth visibility on facial expression processing, we acknowledge certain limitations that merit further consideration. One notable limitation concerns the potential confound of emotional intensity between open- and closed-mouth expressions. Expressions with open mouths may systematically convey greater emotional intensity than their closed-mouth counterparts, particularly for emotions such as happiness and fear. If present, this could complicate the interpretation of our results, as differences in N170 amplitude may reflect intensity effects rather than structural differences in teeth visibility.

Prior research provides support for our interpretation that N170 effects are unlikely to be driven by intensity differences. Shuang et al. (2023) demonstrated that while expression intensity influenced later ERP components such as the vertex positive potential (VPP) and late positive potential (LPP), it did not affect the N170. Similarly, Langeslag et al. (2018) found that open-mouth expressions enhanced early posterior negativity (EPN) amplitudes and subjective emotional responses but primarily influenced later processing stages. These findings suggest that early ERP components like the N170 are more influenced by structural factors, such as teeth visibility, rather than emotional intensity.

We also recognize the inherent difficulty in disentangling the effects of teeth visibility and emotional intensity, as certain expressions (e.g., happiness and fear) inherently involve the exposure of teeth. While this overlap may act as a mediator rather than a confound, future studies using stimulus sets that explicitly control for these variables, such as AI-generated faces, could provide additional clarity. Advanced stimulus control techniques could further elucidate the specific contributions of intensity and structural features to ERP responses.

In conclusion, while the current study provides compelling evidence for the role of teeth visibility in modulating N170, further research is needed to refine our understanding of these effects. By addressing these limitations, future investigations can advance a more nuanced understanding of the neural mechanisms underlying face perception and emotional expression processing.

ADDITION TO ABSTRACT:

While these findings challenge prior research suggesting that N170 is directly influenced by emotional expression, they also highlight the potential role of emotional intensity as an alternative explanation. This underscores the importance of further studies to disentangle these effects.

REVIEWER 2: Minor Issues:

  1. Please include a partial eta-squared for the open vs. closed main effect. This will help gauge if the effect size is meaningful.

ANSWER: The values were added to the table. The small effect sizes were acknowledged in the discussion section; more specifically, the following text was added

It is worth noting that the effect sizes, as measured by partial eta squared for N170 amplitudes, were very small in this study: 0.0026 for mouth position, 0.0016 for emotional expression, and 0.0001 for the interaction between these variables. Despite these small effect sizes, the difference in N170 amplitude for mouth position (closed mouth vs. exposed teeth) was highly significant (p < 0.01). This paradox of small effect sizes alongside significant results raises the possibility that the subtle influence of mouth position on N170 amplitude may have been overlooked in previous studies. This could potentially account for some of the inconsistencies in findings across studies, particularly in research that did not systematically control for the presence of visible teeth. These results underscore the importance of examining even small but statistically robust effects to refine our understanding of face-processing mechanisms.

  1. I see a significant P100 interaction effect described in the results. However, the manuscript does not discuss the potential implications of this finding.  Please include a few lines about the P100 effect and what it may mean. This will also give closure to the reader given the succinct exposition on the role of P100 and N170 components in emotional expression processing in section 1.4. 

ANSWER: The following text was added to the discussion section:

The observed interaction effect between emotional expression and mouth position on P100 amplitude, albeit with a p-value just below 0.05, suggests that the early stages of facial processing may be influenced by the combination of emotional content and specific facial features. This interaction indicates that the P100 component might reflect an early integration of basic visual features and emotional cues, potentially influencing subsequent stages of face processing. While the effect size is modest, this finding underscores the need to further investigate the role of the P100 in processing emotional expressions, particularly in contexts where visual salience is modulated by facial features like teeth visibility.

ADDITIONAL NOTE: In response to a request of Reviewer 1 we reduced the number of electrodes considered for the analysis; consequently, we reran the analysis, slightly affecting the results displayed in Figures 2 and 3; and values displayed in Table 1.

Round 2

Reviewer 2 Report

Comments and Suggestions for Authors

Thanks for providing your responses to my comments. While I am mostly satisfied with the changes made in the newer version, which acknowledges that the study does not control for emotional intensity, I feel that the revision does not provide a balanced argument that can allow the reader to make up their own mind based on the evidence presented. If the writing can be modified slightly to incorporate a more balanced discussion of the matters described below, I would be willing to support publication.

Minor Issue: 

While the N170 is often thought to be associated with structural encoding of faces, the literature has many reports showing that it is influenced by a multitude of factors, emotional intensity also being one of them. For instance, Hinojosa et. al. (2015) using a meta-analysis revealed that the N170 is modulated by emotional intensity. Many studies showing the dependence of N170 on emotional intensity use well-validated datasets such as J3DFD (such as Muller-Bardoff et. al. 2016). However, I understand that Muller-Bardoff et. al. 2016 do not control for mouth opening/teeth visibility. On the other hand, Langeslag et. al (2018) and Cui (2021) don't control for emotional intensity just like the current study. Therefore, there is a need for a well-designed study that can investigate how these factors interact to influence ERP signals. However, the current discussion falls short of providing this balanced perspective, since it only selectively cites articles that support its thesis. If the discussion can be adjusted slightly to incorporate this perspective, the paper will be more impactful. 

Author Response

REVIEWER 2: Thanks for providing your responses to my comments. While I am mostly satisfied with the changes made in the newer version, which acknowledges that the study does not control for emotional intensity, I feel that the revision does not provide a balanced argument that can allow the reader to make up their own mind based on the evidence presented. If the writing can be modified slightly to incorporate a more balanced discussion of the matters described below, I would be willing to support publication.

Minor Issue: 

While the N170 is often thought to be associated with structural encoding of faces, the literature has many reports showing that it is influenced by a multitude of factors, emotional intensity also being one of them. For instance, Hinojosa et. al. (2015) using a meta-analysis revealed that the N170 is modulated by emotional intensity. Many studies showing the dependence of N170 on emotional intensity use well-validated datasets such as J3DFD (such as Muller-Bardoff et. al. 2016). However, I understand that Muller-Bardoff et. al. 2016 do not control for mouth opening/teeth visibility. On the other hand, Langeslag et. al (2018) and Cui (2021) don't control for emotional intensity just like the current study. Therefore, there is a need for a well-designed study that can investigate how these factors interact to influence ERP signals. However, the current discussion falls short of providing this balanced perspective, since it only selectively cites articles that support its thesis. If the discussion can be adjusted slightly to incorporate this perspective, the paper will be more impactful. 

ANSWER: We thank the reviewer for this valuable suggestion. In response, we have revised the paragraph discussing the shortcomings of our study to more explicitly address your concerns. Additionally, we updated the title of the section to better reflect its focus, which now reads: “4.6 The Role of Teeth Visibility and Emotional Intensity in Modulating the N170: Insights, Limitations, and Future Directions.”. Please note that a few sentences at the end of Section 4.1, which have become redundant due to these modifications, were deleted.

Below is the revised section in its entirety:

4.6 The Role of Teeth Visibility and Emotional Intensity in Modulating the N170: Insights, Limitations, and Future Directions

Our study investigated structural factors, such as teeth visibility, and their influence on N170 amplitude. However, we acknowledge the well-documented role of emotional intensity in modulating the N170. For instance, a meta-analysis (Hinojosa et al., 2015) demonstrated that emotional intensity reliably enhances N170 amplitude, particularly for highly arousing expressions like fear or anger. This aligns with findings from studies using well-validated datasets, such as J3DFD (Müller-Bardorff et al., 2016), which reveal that emotional intensity interacts with early face-processing stages. However, these studies often fail to control for structural features like teeth visibility, potentially conflating the effects of emotional intensity with visual salience.

Our findings contribute to this body of work by demonstrating that teeth visibility independently modulates N170 amplitude. However, we did not directly manipulate emotional intensity, which represents a limitation of our study. This oversight is particularly relevant since expressions with open mouths may systematically convey greater emotional intensity than their closed-mouth counterparts. Future research should adopt experimental designs that simultaneously control for both teeth visibility and emotional intensity to better understand their interaction and relative contributions to ERP responses.

Disentangling the effects of teeth visibility and emotional intensity is particularly challenging because certain expressions (e.g., happiness and fear) inherently involve the exposure of teeth. The use of AI-generated faces, designed to explicitly control these variables, could help clarify whether open mouths and exposed teeth act as mediators rather than confounds. Such approaches could further elucidate the specific contributions of emotional intensity and structural features to ERP responses.